# Running Gait Complexity During an Overground, Mass-Participation Five-Kilometre Run

**DOI:** 10.3390/s24227252

**Published:** 2024-11-13

**Authors:** Ben Jones, Ben Heller, Linda van Gelder, Andrew Barnes, Joanna Reeves, Jon Wheat

**Affiliations:** 1Sport and Physical Activity Research Centre, Sheffield Hallam University, Olympic Legacy Park, 2 Old Hall Rd, Sheffield S9 3TY, UK; b.d.jones@shu.ac.uk (B.J.); linda.vangelder@nhs.net (L.v.G.); a.barnes@shu.ac.uk (A.B.); jon.wheat@ntu.ac.uk (J.W.); 2Sport, Performance and Nutrition Research Group, School of Allied Health, Human Services and Sport, La Trobe University, Melbourne, VIC 3083, Australia; 3Public Health and Sport Sciences Department, University of Exeter Medical School, St Luke’s Campus, 79 Heavitree Rd, Exeter EX2 4TH, UK; j.e.reeves@edu.salford.ac.uk; 4Sport and Human Performance Enhancement Research Centre, Nottingham Trent University, Clifton Campus, Nottingham NG11 8NS, UK

**Keywords:** overground running, gait variability, gait complexity, detrended fluctuation analysis (DFA), inertial measurement unit (IMU), gradient

## Abstract

Human locomotion contains innate variability which may provide health insights. Detrended fluctuation analysis (DFA) has been used to quantify the temporal structure of variability for treadmill running, although it has been less commonly applied to uncontrolled overground running. This study aimed to determine how running gait complexity changes in response to gradient and elapsed exercise duration during uncontrolled overground running. Sixty-eight participants completed an overground, mass-participation five-kilometre run (a parkrun). Stride times were recorded using an inertial measurement unit mounted on the distal shank. Data were divided into four consecutive intervals (uphill lap 1, downhill lap 1, uphill lap 2, downhill lap 2). The magnitude (SD) and structure (DFA) of stride time variability were compared across elapsed exercise duration and gradient using a repeated-measures ANOVA. Participants maintained consistent stride times throughout the run. Stride time DFA-α displayed a moderate decrease (*d* = |0.39| ± 0.13) during downhill running compared to uphill running. DFA-α did not change in response to elapsed exercise duration, although a greater stride time SD was found during the first section of lap 1 (*d* = |0.30| ± 0.12). These findings suggest that inter- and intra-run changes in gait complexity should be interpreted in the context of course elevation profiles before conclusions on human health are drawn.

## 1. Introduction

Intra-individual variability is an innate feature of human movement. When a repetitive task is performed, individuals will exhibit movement variability to achieve the same outcome [1]. The variability seen in running gait reflects the ability of the human body to utilise its abundant degrees of freedom whilst satisfying individual, environmental and task constraints [2]. Variability might provide an indicator of a system’s underlying health and protect against overuse injury by providing a wider distribution of loads to structures within the body [3,4]. This seems important since running-related injury (RRI) prevalence is approximately 45% and the majority of RRIs are due to overuse [5].

Movement variability has traditionally been assessed using linear measures such as standard deviation (SD) or coefficient of variation (CV) [1]. Whilst linear measures provide insight into the magnitude of variability, they fail to recognise the temporal dynamics of gait since they consider strides to be independent [6]. Since the mid-1990s, measures have been developed to assess the *complexity* of stride-to-stride variations over time [7,8]. Complexity can be viewed as the “richness” of a signal, and it considers patterns of non-random, structured behaviour in a high-order variable [9]. A high-order variable is a compound of simpler variables and is thought to describe the collective behaviour of a system’s elements [3]. The patterns of behaviour seen in a high-order variable arise from the non-linear physiological interactions between structures within the human body operating at different temporal scales [8,10]. A loss of complexity has been proposed to be associated with injury since any reduction in the body’s degrees of freedom, as evidenced by fewer non-linear interactions, may lead to overuse injury [11]. Reciprocally, an ongoing injury will itself reduce the body’s degrees of freedom. Stride time is a frequent choice of high-order variable in running gait research [6,12], probably due to its relative ease of measurement.

Detrended fluctuation analysis (DFA) is one measure to quantify the complexity of a signal, and it has been widely used in running gait research [13,14,15,16,17,18]. DFA assesses the fractal properties of a signal and returns a scaling exponent, α, quantifying statistical persistence [7]. A scaling exponent 0 < α < 0.5 indicates anti-persistence, whereas a scaling exponent 0.5 < α < 1.0 indicates persistence [7]. Special cases occur at α = 0.5, which indicates that a signal consists of white noise and that any value in a time series is completely uncorrelated to previous values, and at α = 1.0, which indicates that a signal exhibits 1/f behaviour (pink noise) [7]. When α > 1.0, a signal is nonstationary and no longer exhibits a power-law relationship, with α = 1.5 indicating Brownian motion [19].

The monitoring of DFA-α has been proposed as a tool to forecast sports injury [3,18]. For such monitoring to occur in the overground environments where runners typically conduct their training [20], an understanding of the factors that affect complexity measures is required. Evidence for a relationship between the scaling exponent α and elapsed exercise duration remains inconclusive [6], whilst limited evidence from fixed-speed treadmill running has suggested a decrease in DFA-α during downhill running [15]. Wearables, such as inertial measurement units (IMUs), provide the potential for field-based analysis of variability during gait at relatively low cost and place minimal demands on research participants [21]. IMUs, and isolated accelerometers, have been used infrequently to assess the fractal dynamics of gait during overground running, with studies limited to level ground, individual track-based time trials or conducted with low participant numbers [13,14,16]. These studies have found some evidence for a reduction in DFA-α with elapsed exercise duration, although within the constraints of controlled indoor track running [13,14]. There thus remains a need to investigate the effects of gradient and elapsed exercise duration on DFA-α during representative overground running.

The aim of this study was to quantify the effects of gradient and elapsed exercise duration (first half of the run vs. second half of the run) on DFA-α values for stride times collected during an overground, mass-participation five-kilometre run.

## 2. Materials and Methods

### 2.1. Participants & Protocol

The secondary dataset for this study consisted of IMU data recorded from 133 participants (66 female, 67 male; mean ± SD: age 39 ± 12 years; height 1.70 ± 0.09 m; mass 69.2 ± 12.2 kg; run frequency 2.7 ± 1.7 per week; run distance 19.3 ± 15.4 km per week) during an overground, recreational, mass-start five-kilometre run. These data were collected as part of a pre-screening process for a previously published study on peak tibial acceleration which gained ethical approval from Sheffield Hallam University [22].

Briefly, participants were recruited to run at Endcliffe parkrun (https://www.parkrun.org.uk/endcliffe; accessed on 12 April 2024). Parkrun is a free, weekly, timed five-kilometre run that takes place worldwide. Over 9 million people had registered to complete a parkrun at the end of 2023 [23]. At the time of data collection, the Endcliffe parkrun course comprised two laps on a smooth tarmac surface, each consisting of an uphill section (+2.3% gradient) followed by a downhill section (−2.0%) of approximately equal length, partitioned by a sharp turn. Participants were asked to run the course as normal with a RunScribe™ IMU (RunScribe version 2, Scribe Labs, Half Moon Bay, CA, USA) attached to the anteromedial aspect of the right tibia, five centimetres above the medial malleolus [24,25]. The accelerometer (±16 g) within the IMU sampled at 500 Hz, started recording when a threshold of 3 g was reached, and stopped recording when the signal dropped below this threshold for 15 s.

### 2.2. Data Processing

Data from the RunScribe™ IMUs were imported to MATLAB R2023a (MathWorks, Natick, MA, USA) for further processing. Accelerometer data were filtered using a fourth-order low-pass Butterworth filter with a cutoff frequency of 50 Hz and the offset removed. In some instances, the signal saturated at 16 g. Therefore, rather than using the time between peaks in vertical acceleration, our data were linearly interpolated, and stride times were calculated using a zero-threshold crossing method, which has been shown to accurately detect gait events for shank-mounted accelerometers [26]. Using this zero-threshold crossing method, peaks in vertical acceleration with a minimum height of 2 g and separated by a minimum of 500 ms were identified. Foot strike was then defined as the time at which the vertical acceleration signal crossed a threshold of 0 g immediately prior to peak vertical acceleration. Stride time was calculated as the difference between consecutive foot strikes.

Participants were likely to have walked around before and after the five-kilometre run. Within the first 100 strides, the last stride time longer than the mean plus two standard deviations was found. This stride time defined the start of the run. The same algorithm was applied to the final 50 strides to define the end of the run. Participants may also have walked during the five-kilometre run. Within the trimmed stride time series, strides that took longer than the mean plus four standard deviations were considered to be walking and were flagged in the data. The course was then split into four sections (uphill lap 1, downhill lap 1, uphill lap 2, downhill lap 2) based on sharp turns in the run route that were identified using the gyroscope signal.

A secondary filtering procedure adopting a more conservative approach to participant inclusion was then applied. Stride times were flagged if they corresponded to step frequencies lower than 130 steps per minute (spm), greater than 210 spm [27,28], or if they changed by more than 0.11 s between consecutive strides, calculated as 30% of the difference between the maximum and minimum permitted stride times.

The accuracy of fractal analysis is dependent on the number of included data points [29]. We chose to analyse 256 strides from each run section as a compromise between the inclusion of the greatest number of participants and the minimum number of strides for a reliable estimate of fractal measures [30]. Participants were included in the final analysis when they met the criteria of 256 consecutive running strides for all four sections (68 participants; 51% of the initial cohort; 37 female, 31 male; mean ± SD: age 38 ± 11 years; height 1.72 ± 0.10 m; mass 69.5 ± 14.9 kg; run frequency 3.1 ± 1.4 per week; run distance 21.3 ± 15.2 km per week). When a participant had more than 256 running strides during a section, the first 256 strides were taken for analysis.

The mean and SD of stride times were then found. The evenly spaced averaged DFA algorithm [31], which provides reliable results with short stride time series [32], was implemented using the functions presented by Liddy and Haddad [32]. Parameters were selected based on an a priori analysis of simulated fractional Brownian motion (fBm) and fractional Gaussian noise (fGn), generated using the algorithm proposed by Kroese and Botev [33]. Specifically, 100 time series of length N = 256 were generated for each of 11 values of α, ranging from 0.3 to 1.4 in steps of 0.1. The value of 1.0 was excluded due to the discontinuity between fBm and fGn [34]. This range covered a conservative estimate of expected α values from running gait [12]. The parameters that exhibited the smallest root mean square error between theoretical and estimated values of alpha were used in the subsequent analysis. Window sizes ranged from 4 to 64, and a polynomial of order 1 was used to detrend each window.

### 2.3. Statistical Analysis

Statistical analyses were performed in MATLAB R2023a. Kolmogorov–Smirnov tests were used to confirm the normality of the data. A 2 × 2 repeated measures ANOVA was used to compare uphill and downhill sections (gradient) and laps 1 and 2 (elapsed exercise duration) for each measure of variability. The magnitude of any difference was estimated as the Cohen’s *d* effect size. Values of 0.2, 0.5 and 0.8 were identified as the lower thresholds for small, moderate and large effects, respectively [35]. Statistical significance was set at *p* ≤ 0.05.

## 3. Results

The average time to complete the five-kilometre run was 25.5 ± 4.7 min. The mean, SD and DFA-α of stride times during each section of the run are displayed in Figure 1, Figure 2 and Figure 3, respectively. Stride times did not change over the course of the run.

Stride time DFA-α displayed a significant main effect for gradient (F(1, 67) = 22.5, *p* < 0.001), with a moderate decrease (*d* = |0.39| ± 0.13; uphill = 0.75 ± 0.15, downhill = 0.68 ± 0.16) found during downhill running compared to uphill running. However, elapsed exercise duration did not have a significant effect on stride time DFA-α (F(1, 67) = 2.97, *p* = 0.09), with only a small decrease (*d* = |0.11| ± 0.12; lap 1 = 0.72 ± 0.16, lap 2 = 0.70 ± 0.15) occurring from lap 1 to lap 2 of the run. The gradient by elapsed exercise duration interaction was also not significant (F(1, 67) = 0.11, *p* = 0.74).

A significant elapsed exercise duration by gradient interaction (F(1, 67) = 24.5, *p* = 0.001) was found for stride time SD. Stride time SD was lower (moderate effect size: *d* = |0.30| ± 0.12) during downhill (13.30 ± 3.07 ms) than uphill (14.40 ± 3.68 ms) running on lap 1 but similar on lap 2 (*d* = |0.03| ± 0.12). Stride time SD displayed a significant main effect for elapsed exercise duration (F(1, 67) = 16.8, *p* < 0.001), with a small decrease between lap 1 and lap 2 (*d* = |0.26| ± 0.12; lap 1 = 13.85 ± 3.42 ms, lap 2 = 12.89 ± 2.99 ms), but no significant main effect for gradient.

## 4. Discussion

The purpose of this study was to quantify the effects of gradient and elapsed exercise duration on DFA-α values for stride times collected during an overground, mass-participation five-kilometre run. Across all participants and conditions, a mean DFA-α of 0.71 was calculated. This statistical persistence in stride times is a common finding in running gait research [12]. The DFA-α of stride times displayed a significant decrease during downhill running compared to uphill running but no significant effect for elapsed exercise duration.

A downhill running section induced a moderate (−10%) reduction in DFA-α compared to an uphill section. The DFA-α values found here during downhill running were typically lower than those reported in studies on healthy participants for both treadmill [36,37,38] and overground [13,16,38] running on level surfaces. Although running speed was not measured, downhill running was likely characterised by greater running speeds and longer stride lengths, which may have been less adaptive and flexible [15]. A reduction in DFA-α during downhill treadmill running compared to uphill treadmill running at the same fixed speed has previously been observed with an optoelectronic device (0.58 and 0.72 at −2% and +2% gradients respectively), although the same effect was not found when measuring stride times with an IMU [15]. Despite the fact that we collected data from an overground mass-participation run with no control over speed or exertion level, we also found a reduction in DFA-α in downhill running. It may be that a fear of slipping or falling during downhill running leads to runners adopting a more “cautious” gait which exhibits less persistence, as has been proposed in individuals with central nervous disease [39]. Reductions in DFA-α have also been observed between previously injured (0.79) and uninjured groups (0.96) [13], which may suggest that the constraints imposed by downhill running induce a similar, although lesser, reduction in running gait complexity as those imposed by previous running-related injury. Nevertheless, inter- and intra-run differences in gradient should be considered before conclusions on running-related injury are drawn from changes in gait complexity. Given the non-linear relationship between DFA-α and running speed [36,38], follow up research that explores the relationship between gradient and the DFA-α of stride times and stride lengths during overground running appears warranted.

DFA-α exhibited no significant effect for elapsed exercise duration, although the SD of stride times decreased significantly during lap 2 of the run. Elapsed exercise duration has been associated with fatigue during studies on running gait complexity [13,14,16], with fatigue being proposed to limit the ability of the human body to utilise its degrees of freedom, thus reducing running gait complexity and potentially increasing injury risk [4]. The representativeness of this study and the non-competitive ethos of parkrun events meant that individual runners may have been running conservatively and thus any links between elapsed exercise duration and fatigue are speculative. Published evidence of a clear relationship between fatigue and the DFA-α of stride times remains inconclusive, with studies finding a reduction in DFA-α with fatigue [13,14], a non-linear relationship between DFA-α and fatigue [17] and no relationship between DFA-α and fatigue [16]. However, the vast range of fatiguing protocols [14,16,17], the habitual lack of measurement of a direct marker of fatigue [6] and differences in time series length [29] make comparisons between studies challenging. The DFA-α values observed here likely better reflect the changes in gait complexity that runners experience during a typical training run as opposed to a flat-out race or time-trial.

Stride times remained consistent throughout all sections of the run. Relative consistency in stride times for an individual may minimise their metabolic cost [40], as has been observed during runs over flat overground courses of a comparable distance [13,14,41] and during runs over overground courses with similar topological profiles [42]. Therefore, any changes in running speed with gradient and elapsed exercise duration would need to have been regulated by stride length. Regarding stride time variability, the significant elapsed exercise duration by gradient interaction was a result of the increased magnitude of variability seen in the uphill section of lap 1. This may represent a genuine interaction between elapsed time and gradient, or it may reflect changes in stride times caused by greater congestion at this stage of the run which induced increases in the magnitude of variability but had no effect on DFA-α.

In this study, data were collected from a popular, overground, mass-participation run, so a strength is that we have demonstrated the effect of gradient and elapsed exercise duration in a widely representative environment. However, the extent to which these isolated effects can be interpreted in other contexts, such as laboratory-based studies, is limited. Factors that should be considered include: (1) congestion varied throughout the course of the run; (2) uphill and downhill running were completed on independent course sections that each possessed different corners, path widths, street furniture and subtle variations in smooth tarmac surface; (3) running speed and perceived and physiological exertion were not controlled; (4) analysis focused on the DFA-α of running stride times and thus only included participants for whom we could construct series of length N = 256 for running stride times.

## 5. Conclusions

This study was the first to apply DFA to stride times measured in an environment representative of a typical uncontrolled overground run. DFA-α values of stride times were found to decrease during downhill running compared to uphill running but did not change with elapsed exercise duration. Inter- and intra-run changes in running gait complexity should therefore be interpreted in the context of course elevation profiles before conclusions on the health of the human system are drawn. Future studies should investigate other environmental and task constraints to better understand the clinical implications of changes to running gait complexity during overground running.

## Figures and Tables

**Figure 1 sensors-24-07252-f001:**
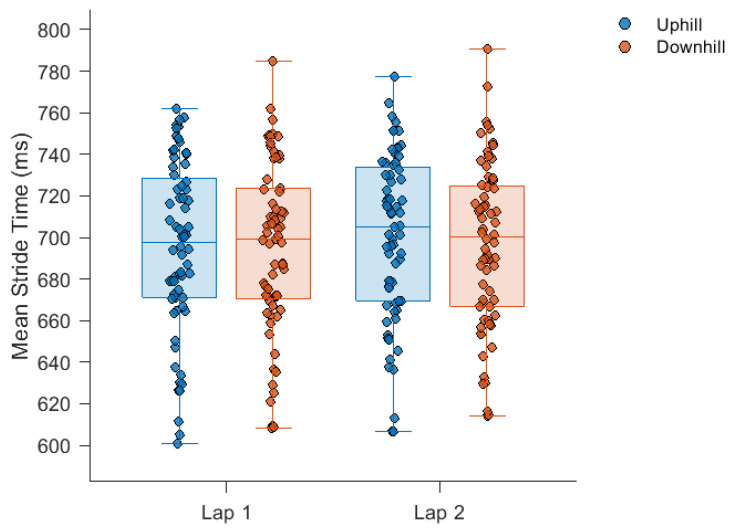
Stride times during each section of the run. No significant main effects were observed. The box plots illustrate, from bottom to top, the minimum, first quartile, median, third quartile and maximum value in each course section.

**Figure 2 sensors-24-07252-f002:**
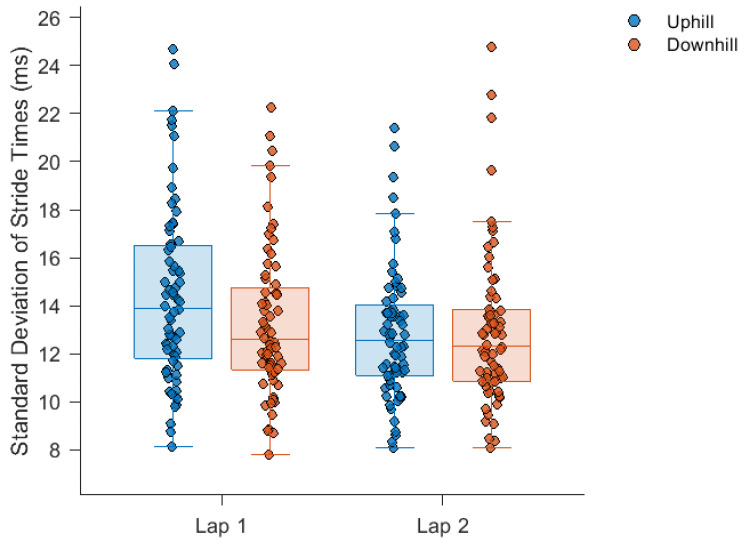
Stride time standard deviation during each section of the run. Significant main effect for elapsed exercise duration (lap 1 vs. lap 2), *p* < 0.001. Significant elapsed exercise duration by gradient interaction, *p* = 0.001. The box plots illustrate, from bottom to top, the minimum, first quartile, median, third quartile and maximum value in each course section.

**Figure 3 sensors-24-07252-f003:**
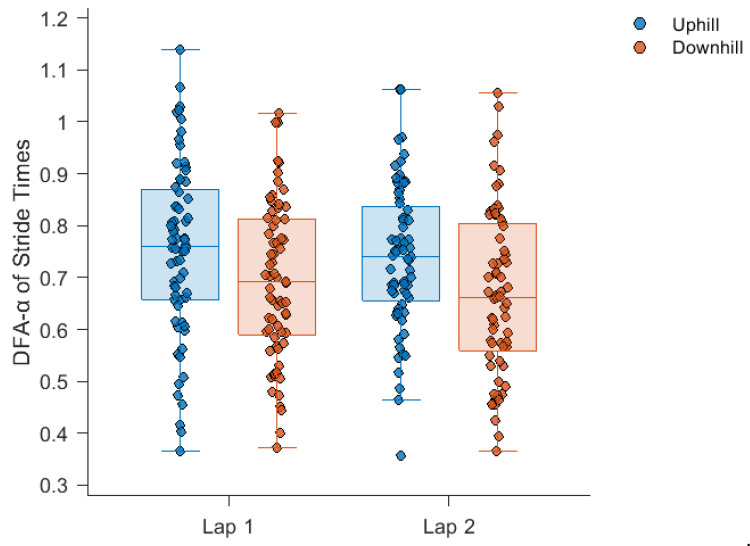
Stride time DFA-α during each section of the run. Significant main effect for gradient (uphill vs. downhill), *p* < 0.001. The box plots illustrate, from bottom to top, the minimum, first quartile, median, third quartile and maximum value in each course section.

## Data Availability

The data presented in this study are available on request from the corresponding author. The data are not publicly available due to ethical restrictions.

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
