# Peer review of "Running Gait Complexity During an Overground, Mass-Participation Five-Kilometre Run"

_sensors, 2024, doi:10.3390/s24227252_

Round 1
Reviewer 1 Report
Comments and Suggestions for Authors
This study apply DFA to stride times measured in an environment representative of a typical uncontrolled overground run is interesting. But there are also some issues:
1. Participant Characteristics: Include comprehensive information about the participants, such as experience level, to facilitate comparison with future studies.
2. Cautious Interpretation: Participants maintained consistent stride times throughout the uncontrolled overground run is important or not for running economy and injury risk。
3. Data Sharing: This study only show the data of “stride time”,plelease consider sharing more raw data or a portion of it, along with detailed data processing and analysis methods. This allows other researchers to verify the findings and conduct further analyses.
4. Expanding on Practical Applications: Discuss how the findings could be applied in reduce running-related injury.
5. Detailed Methodology: Provide a more detailed description of the uncontrolled overground,such as road surface,and four consecutive intervals。
6. Line 99-100: “[24]”maybe need new referrence.
7. Text editing: Line 352: “32. D.P. Kroese, Z.I. Botev, Spatial Process Generation, (2013). https://doi.org/10.48550/arXiv.1308.0399. ”
Reviewer 2 Report
Comments and Suggestions for Authors
This manuscript presents an interesting application of IMU and DFA for field-based running. It is well written and informative throughout. The methods can be improved by providing reliability information and greater clarity about the data removed from analysis. The lack of participant characteristics and contextual information about the 5km run limit the generalisability and insight that can be gained from the results. I have provided some specific comments below that I hope are helpful.
Abstract - Please add effect magnitude statistics with confidence intervals to results
Introduction is clear and informative.
Methods
2.1. Line #91: This reference is to an intervention study with 11 participants. Can you make it clearer how that relates to this dataset of 133 participants?
2.1. Line #87-89: Please add more descriptive information i.e., weekly running distance, 5km time, etc. These descriptives are reported in the primary study you cite so they should be available to report here. This descriptive information is important to help with generalisability of your results and, as I allude to later, can help look for moderators to your results.
2.2. Line #105-115: please report reliability of stride time measurements using your method.
2.2. Line #138-140: please report reliability of DFA alpha measurements for stride times using your method
2.2. Line #116-129: Please report if the volume of data removed was different or not for uphill and downhill portions of the course.
Results
Line #159: Please provide participant characteristics to support generalisability of your findings i.e., characteristics for the final set of participants and include the descriptives for weekly running, 5km time, etc. mentioned above.
Line #162-175: Without any data on perceived or physiological exertion it is difficult to interpret the exercise duration effect and interaction i.e., there were likely participants that just cruised through the 5K and others who pushed their limits. This issue would be helped by exploring the impact of runner experience and/or ability on exercise duration findings i.e., in addition to adding descriptive information for participant characteristics, the running experience and ability characteristics should be explored as moderating variables. E.g., include 5K competition time as a covariate or subgroup 5K time into running ability categories. Please see my comment in the methods about the descriptive information that you should have for these runners based on the primary study the data are reported to have come from i.e., weekly running distance and 5km time are reported in that study. Integrating those factors here in your results would provide more insight to your statistical findings i.e., do your findings persists when running experience/volume/5km time/etc are considered as moderators? If the answer is yes or no it helps add more context to your results and is informative to the subsequent speculation you include in your discussion.
Line #162-175: Please add confidence intervals for all Cohen’s d values.
Discussion
Line #209-211: please elaborate here regarding how a reduction in alpha reflects having greater control over stride time. This seems counterintuitive if alpha 0.5 is random variation.
Line #215-230: This speculation would be improved greatly if you can contextualise your analysis with the participant descriptives and moderators highlighted above.
Line #235-239: discuss whether this same issue would have impacted DFA alpha findings.
Line #240-246: the lack of contextual data should be highlighted here as an important limitation i.e., lack of pace data for each run section as well as measures of perceptual and physiological exertion. Without pace data it is unclear if gradient or speed differences caused the study findings and without exertion data it is unclear what the exercise duration reflects (e.g., was it a “warming-up” run at low exertion or was it a fatiguing high-exertion effort). Additionally, dropping 50% of participants from the analysis reduces the generalisability to many parkrun participants and that should be acknowledged.
Round 2
Reviewer 1 Report
Comments and Suggestions for Authors
Accept in present form
Reviewer 2 Report
Comments and Suggestions for Authors
Thank you for your responses and revisions. I have no further suggestions.